# Adjuvant Radiotherapy in Patients with Squamous Cell Carcinoma of the Oral Cavity or Oropharynx and Solitary Ipsilateral Lymph Node Metastasis (pN1)—A Prospective Multicentric Cohort Study

**DOI:** 10.3390/cancers15061833

**Published:** 2023-03-18

**Authors:** Peer W. Kämmerer, Silke Tribius, Lena Cohrs, Gabriel Engler, Tobias Ettl, Kolja Freier, Bernhard Frerich, Shahram Ghanaati, Martin Gosau, Dominik Haim, Stefan Hartmann, Max Heiland, Manuel Herbst, Sebastian Hoefert, Jürgen Hoffmann, Frank Hölzle, Hans-Peter Howaldt, Kilian Kreutzer, Henry Leonhardt, Rainer Lutz, Maximilian Moergel, Ali Modabber, Andreas Neff, Sebastian Pietzka, Andrea Rau, Torsten E. Reichert, Ralf Smeets, Christoph Sproll, Daniel Steller, Jörg Wiltfang, Klaus-Dietrich Wolff, Kai Kronfeld, Bilal Al-Nawas

**Affiliations:** 1Department of Oral and Maxillofacial Surgery—Plastic Operations, University Medical Center Mainz, Augustusplatz 2, 55131 Mainz, Germany; 2Hermann-Holthusen-Institute for Radiation Oncology, Asklepios Hospital St. Georg, University Medical Center Hamburg-Eppendorf, 20246 Hamburg, Germany; 3Department of Oral & Cranio-Maxillofacial Surgery, University Hospital of Luebeck, Ratzeburger Allee 160, 23538 Lübeck, Germany; 4Department of Oral and Craniomaxillofacial Plastic Surgery, University of Giessen and Marburg, UKGM, Campus Marburg, Faculty of Medicine, Philipps-University of Marburg, Baldingerstrasse, 35043 Marburg, Germany; 5Department of Cranio- and Maxillofacial Surgery, Hospital of the University of Regensburg, Franz-Josef-Strauß-Allee 11, 93053 Regensburg, Germany; 6Department of Oral and Maxillofacial Surgery, Saarland University Medical Center, Kirrberger Str., 66424 Homburg, Germany; 7Department of Oral and Maxillofacial Surgery, Facial Plastic Surgery, Rostock University Medical Center, Schillingallee 35, 18057 Rostock, Germany; 8Clinic for Maxillofacial and Plastic Surgery, University Hospital Frankfurt, Goethe University, Marienburgstraße 2, 60528 Frankfurt am Main, Germany; 9Department of Oral and Maxillofacial Surgery, University Medical Center Hamburg-Eppendorf, Martinistraße 52, 20251 Hamburg, Germany; 10Department of Oral and Maxillofacial Surgery, University Hospital Carl Gustav Carus, Fetscherstr. 74, 01307 Dresden, Germany; 11Department of Oral and Maxillofacial Plastic Surgery, University Hospital Würzburg, Pleicherwall 2, 97070 Würzburg, Germany; 12Department of Oral and Maxillofacial Surgery, Charité—Universitätsmedizin Berlin, Corporate Member of Freie Universität Berlin, Humboldt-Universität zu Berlin, and Berlin Institute of Health, Augustenburger Platz 1, 13353 Berlin, Germany; 13Institute of Medical Biostatistics, Epidemiology and Informatics, University Medical Center of the Johannes Gutenberg-University Mainz, Rhabanusstraße 4, 55118 Mainz, Germany; 14Department of Oral and Maxillofacial Surgery, University Hospital of Tübingen, Osianderstraße 2-8, 72076 Tübingen, Germany; 15Department of Oral and Maxillofacial Surgery, University Hospital Heidelberg, Im Neuenheimer Feld 400, 69120 Heidelberg, Germany; 16Department of Oral and Maxillofacial Surgery, University Hospital of Aachen, Pauwelsstraße 30, 52074 Aachen, Germany; 17Department of Oral and Maxillofacial Surgery, Justus-Liebig-University, Klinikstraße 33, 35392 Gießen, Germany; 18Department of Oral and Maxillofacial Surgery, Friedrich-Alexander University Erlangen-Nürnberg, Glückstraße 11, 91054 Erlangen, Germany; 19Department of Oral and Maxillofacial Surgery, University Hospital Ulm, Albert-Einstein-Allee 11, 89081 Ulm, Germany; 20Department of Oral and Maxillofacial Surgery/Plastic Surgery, University Medicine Greifswald, Fleischmannstraße 8, 17489 Greifswald, Germany; 21Department of Oral and Maxillofacial Surgery, Heinrich Heine University Düsseldorf, Moorenstraße 5, 40225 Düsseldorf, Germany; 22Department of Oral and Maxillofacial Surgery, University Hospital of Schleswig-Holstein, Campus Kiel, Arnold-Heller-Straße 3, 24105 Kiel, Germany; 23Department of Oro-Maxillofacial Surgery, University Hospital Rechts der Isar, Technical University of Munich, Langerstraße 3, 81675 München, Germany; 24Interdisciplinary Center for Clinical Trials, University Medical Center of the Johannes Gutenberg University, Langenbeckstraße 1, 55131 Mainz, Germany

**Keywords:** oral squamous cell carcinoma, oropharyngeal carcinoma, surgery, resection, radiotherapy, survival, progression-free survival, quality of life, prospective, multicentric, lymph node, pN1

## Abstract

**Simple Summary:**

In brief, this is the first prospective study using a multicentric approach to investigate the effectiveness of adjuvant radiotherapy in patients with early squamous cell carcinoma of the oral cavity (T1/2) and the oropharynx (T1) with one single ipsilateral cervical lymph node metastasis (pN1) in terms of overall survival, time to progression, and quality of life. After the inclusion of 209 patients into this prospective multicentric comprehensive cohort study (2009–2021) and analyzing the follow-up data, we can conclude that adjuvant radiotherapy in patients with early squamous cell carcinoma of the oral cavity and oropharynx does not seem to influence overall survival, but it positively affects the time to progression. However, irradiated patients report a significantly decreased quality of life up to three years after therapy compared to the observation group.

**Abstract:**

(1) Background: Evaluation of impact of adjuvant radiation therapy (RT) in patients with oral squamous cell carcinoma of the oral cavity/oropharynx (OSCC) of up to 4 cm (pT1/pT2) and solitary ipsilateral lymph node metastasis (pN1). A non-irradiated group with clinical follow-up was chosen for control, and survival and quality of life (QL) were compared; (2) Methods: This prospective multicentric comprehensive cohort study included patients with resected OSCC (pT1/pT2, pN1, and cM0) who were allocated into adjuvant radiation therapy (RT) or observation. The primary endpoint was overall survival. Secondary endpoints were progression-free survival and QL after surgery; (3) Results: Out of 27 centers, 209 patients were enrolled with a median follow-up of 3.4 years. An amount of 137 patients were in the observation arm, and 72 received adjuvant irradiation. Overall survival did not differ between groups (hazard ratio (HR) 0.98 [0.55–1.73], *p* = 0.94). There were fewer neck metastases (HR 0.34 [0.15–0.77]; *p* = 0.01), as well as fewer local recurrences (HR 0.41 [0.19–0.89]; *p* = 0.02) under adjuvant RT. For QL, irradiated patients showed higher values for the symptom scale pain after 0.5, two, and three years (all *p* < 0.05). After six months and three years, irradiated patients reported higher symptom burdens (impaired swallowing, speech, as well as teeth-related problems (all *p* < 0.05)). Patients in the RT group had significantly more problems with mouth opening after six months, one, and two years (*p* < 0.05); (4) Conclusions: Adjuvant RT in patients with early SCC of the oral cavity and oropharynx does not seem to influence overall survival, but it positively affects progression-free survival. However, irradiated patients report a significantly decreased QL up to three years after therapy compared to the observation group.

## 1. Introduction

In squamous cell carcinoma of the oral cavity and oropharynx (OSCC), lymph node metastases considerably affect patients’ prognosis. Resection of the primary tumor, together with appropriate neck dissection, is considered to be the therapeutic gold standard [1]. It was shown that about one-third of OSCC patients experience recurrent locoregional disease and distant metastases [2]. Number, site, and size of affected lymph nodes, extranodal extension (ENE), and lymph node ratio are described to be prognostic factors for overall survival [3,4]. Interdisciplinary guidelines on OSCC recommend risk-adapted adjuvant radio(chemo)therapy (R(C)T) in cases with ENE, R1-resection, multiple nodal metastases, and/or contralateral lymph node involvement (pN > 1), as well as invasion of blood (V1) and/or lymphatic vessels (L1) [5,6]. For OSCC with a diameter of less than 4 cm in the greatest dimension (pT1/T2) with only one single lymph node metastasis without risk factors and distant metastases (cM0), there is no clear recommendation for adjuvant radiotherapy (RT) [5,7], and the decision upon adjuvant RT is mainly based on institutional and/or patients’ preferences. Clinical Practice Guidelines of the American Society of Clinical Oncology (ASCO) do not recommend adjuvant neck radiotherapy to patients with a single pathologically positive node (pT1) without extranodal extension unless there are other indications, such as perineural invasion, lymphovascular space invasion, or a T3/4 primary [6]. In accordance, the low risk for regional spread of these OSCCs might be treated with less intense therapy, sparing the affected patients from significant additional treatment-related toxicity.

While some authors did not find any evidence for survival or locoregional benefit in patients with pN1 treated with adjuvant RT [8,9], others report better survival and locoregional control with adjuvant RT [10,11,12,13,14]. However, the published studies must be interpreted cautiously because they are mostly retrospective, have small patient numbers, various endpoints, and a heterogenous patient population, and they lack consistent inclusion criteria. In retrospective studies, patients who received surgery followed by RT generally have more advanced disease and adverse pathological features [12].

Nevertheless, quality of life (QOL) of irradiated patients with pT1/2 pN1 pM0 OSCC has yet to be evaluated prospectively and was not compared to a control group yet. This is of utmost importance, as those data will reflect the patient’s experience with the disease, treatment, or changes in their sense of wellbeing. Many patients will even consider the effect of adjuvant RT on their QOL to be as significant as the effect of RT in terms of the chance of cure [15]. In particular, surgery-related side effects might be even more aggravated by the addition of adjuvant RT. In conclusion, if it would be possible to omit adjuvant RT in this highly selective patient population while maintaining excellent overall survival and locoregional control, the potential benefit for QOL should be a significant factor in decision-making.

Therefore, this study aimed to investigate the effectiveness of adjuvant RT in patients with early squamous cell carcinoma of the oral cavity (T1/2) and the oropharynx (T1) with one single ipsilateral cervical lymph node metastasis (pN1) in terms of overall survival, progression-free survival, and quality of life.

## 2. Materials and Methods

### 2.1. Patients

In this prospective multicentric study (trial registration: ClinicalTrials.gov: NCT00964977; study approval of the Ethics Committee of the State Medical Council of Rhineland-Palatinate, Germany 837.148.03 (3810)), patients with completely resected (R0) histologically proven squamous cell carcinoma of the oral cavity up to a histologically determined tumor size of a maximum of 4 cm (T1/T2) and the oropharynx up to a histologically determined tumor size of a maximum of 2 cm (T1) were included, regardless of the histological degree of differentiation.

In brief, the tumor had to be located at the anterior 2/3 of the tongue, the bottom of the tongue, the floor of the mouth, the retromolar region, the buccal mucosa, as well as the vallecula, tonsils, tonsillar fossa, palatine arches, glossotonsillar fossa, the posterior pharyngeal wall, the soft palate, or the uvula. There had to be a histologically proven solitary ipsilateral lymph node metastasis < 3 cm (pN1) without ENE and lymphovascular space invasion (L0) following the 7th edition of the TNM staging system (American Joint Committee on Cancer (AJCC)). Eligible patients had to show a general condition that allowed radiotherapy (Karnofsky Performance Status (KPS) ≥ 50% or ECOG ≤ 2). Written informed consent was obtained from all patients before study entry. Patients of <18 years of age, pregnant women, patients with reported drug abuse or intake of substances with a potential to influence compliance or impaired judgment, as well as patients with familial or job-related responsibilities, which might preclude from maintaining the study schedule, were ineligible [16].

### 2.2. Surgical Treatment

In brief, complete resection with a margin of at least 0.5 cm was required during surgery. After tumor resection, primary closure or flap reconstructions were performed according to the defect size. As published by Moergel et al. [16] and per DÖSAK-criteria [17], lymph node resection was oriented in relation to the tumor center, and the classification of cervical regions followed the recommendations of Robbins from 2002 [18].

### 2.3. Allocation to Study Groups

After surgery and histological assessment, included patients were allocated into the two study arms (RT versus observation). The observation group underwent clinical follow-up per protocol. Randomization was conducted one-to-one, with recruitment extending over four years, and ongoing follow-up (Figure 1) continued until the end of the trial. Assuming exponentially distributed survival of 45% within the control group and 55% within the RT group after five years and a drop-out rate of 5% per year, 280 patients per group were required to detect a difference of 5% in overall survival with a power of 70% [16]. Patients who did not agree to randomization, but preferred one of the two arms, were included as prospective observations after giving informed consent. Additionally, if patients did not agree to randomization and could not decide which therapy arm to choose, they were allocated according to the attending physician.

### 2.4. Adjuvant Radiation Therapy

Radiotherapy was scheduled to begin within six weeks after the last surgical intervention with a minimal post-operative healing period of at least eight days. Patients were immobilized using a custom-made face mask. Adjuvant radiation therapy was prescribed, recorded, and reported according to the ICRU 50 [19]. The CTV (clinical target volume) was defined as the primary tumor bed. A 2 cm margin was added to comprise the PTV (planning target volume). Conventional fractionation was used for the primary tumor bed and for the level of the positive neck node with 1.8 Gy/fraction, five days a week to a total dose of 59.4 Gy. Elective nodal irradiation was applied with 1.8 Gy/fraction, five times per week, for a total dose of 50.4 Gy. The decision to treat the contralateral neck was left to the discretion of the radiation oncologist. In general, only ipsilateral neck RT was used for a strongly lateralized tumor. For tumors reaching or crossing the midline, bilateral neck RT was prescribed. For oral cavity tumors, the involved neck was treated from level I through IV with a boost to the involved level, as well as the negative neck from level I-III. Only ipsilateral neck RT (level II-IV) was prescribed for small, lateralized tonsil lesions. In all other oropharyngeal subsites, bilateral neck RT was given (vallecula, base of tongue, posterior pharyngeal wall).

### 2.5. Follow-Up

After the recruitment phase of four years, clinical follow-up included physical examinations, ultrasonography, computed tomography, and/or magnetic resonance imaging, as well as quality of life questionnaires (European Organization for Research and Treatment of Cancer (EORTC) Quality of Life Questionnaire (QLQ-C30 with H&N 35 module); Figure 1). Further biopsy was performed if there was clinical or radiographical evidence of a local recurrence, lymph node, and/or distant metastases. In cases of recurrence, the patients were further treated in accordance with the tumor classification, and they were censored for further examinations.

### 2.6. Study Parameter

#### 2.6.1. Primary Endpoint

The primary endpoint was overall survival measured from the day of surgery.

#### 2.6.2. Secondary Endpoints

##### Progression-Free Survival

Progression-free survival was defined as the absence of lymph node metastases, histologically confirmed local recurrence, and distant metastases during the observation period starting after surgery.

### 2.7. Quality of Life

For assessment of quality of life, the European Organization for Research and Treatment of Cancer (EORTC) Quality of Life Questionnaire (QLQ-C30 with H&N 35 module) was used after six months, as well as one, two, and at least up to three years after the first surgical intervention (Figure 1).

### 2.8. Statistics

The influence of several risk factors (adjuvant radiotherapy (yes/no), neck dissection in accordance to DÖSAK-criteria (see above; yes/no), age, ECOG status, minimum resection margin (R0 < 5mm/R0 ≥ 5mm) tumor stage (T1/T2), and grading (G1 & 2/G3)) were analyzed descriptively, as well as in a multivariate Cox regression proportional hazard model on overall survival, progression-free survival, the occurrence of lymph node metastases, the occurrence of distant metastases, the occurrence of local recurrence, and tumor-related death. For EORTC QLQ-H&N 35, data obtained from the start point of therapy and after 0.5, one, two, and three years were assessed and compared between groups. For all analyses, a *p* < 0.05 was termed to be significant.

## 3. Results

A total of 209 patients from 27 centers with a median observation period of 3.4 years (min: 0.06 years, max: 9.99 years) were included from September 2009 to July 2016 and followed up until the beginning of 2021. An amount of 66% of patients were male (n = 138), and 34% (n = 71) were female, with a mean age of 60 years (standard deviation (SD): 11). Locations of the tumors were pre-canine in 14.4% (n = 30), post-canine in 52.6% (n = 110), and oropharyngeal in 29.7% (n = 62) of cases. In seven cases, the location information was missing. In all patients, an ipsilateral neck dissection was performed.

### 3.1. Patient Characteristics

Of 209 patients, 137 (65.6%) were not irradiated, whereas 72 (34.4%) received adjuvant irradiation. There was no significant difference between gender (*p* = 0.868), age (*p* = 0.217), and tumor site (*p* = 0.728) between the two groups. Patient and treatment characteristics are summarized in Table 1. Patients with pT > 1 and G > 2 were significantly more often irradiated. The study was interpreted as observational, as 175 patients were treated according to preference.

### 3.2. Primary Endpoint

#### 3.2.1. Overall Survival

##### Overall Survival and Irradiation

In total, 38/137 patients (27.7%) in the non-irradiated and 21/72 patients (29.2%) in the irradiated group died. The primary outcome measure showed no difference in overall survival after five years between the two treatment groups (hazard ratio (HR) 0.98 [0.55–1.73], *p* = 0.94; Figure 2). The post hoc calculation showed a power of 16.7% for this assumption.

#### 3.2.2. Other Factors Significantly Influencing Overall Survival

##### ECOG Status

At baseline, 125 patients presented ECOG 0, 67 ECOG 1, and 15 ECOG 2. Two patients were censored due to missing information. Of the included cases, 33/125 (26.4%), 18/67 (26.9%), and 8/15 (53.3%) died. Patients with ECOG 2 showed a significantly decreased survival compared to ECOG 0 (HR 2.45 [1.1–5.47]; *p* = 0.03).

##### Tumor Size

One hundred ten of the included patients (52.6%) had a tumor size of <2 cm (pT1), and 99 patients (47.4%) had a tumor size of 2–4 cm (pT2). Regarding overall survival, 25/110 (22.7%) and 34/99 patients (34.3%) died. Here, pT > 1 showed a significantly decreased overall survival (HR 1.89 [1.1–3.23]; *p*-value: 0.02).

### 3.3. Secondary Endpoints

#### 3.3.1. Progression-Free Survival and Irradiation

An amount of 56/134 in the non-irradiated (41.8%) and 15/66 patients (22.7%) in the irradiated group had an event within five years after surgery (HR 0.37 [0.2–0.69]; *p*-value: <0.001; Figure 3). Here, the post hoc calculation showed a power of 91.92%. In particular, fewer lymphatic node metastases were seen in the irradiated group (non-irradiated 33/132 (25%) versus irradiated 8/66 (12.1%); HR 0.34 [0.15–0.77]; *p*-value: 0.01). There were also significantly more local recurrent diseases in the non-irradiated group (37/134 (27.6%) versus 10/66 (15.2%); HR 0.41 [0.19–0.89]; *p*-value: 0.02).

#### 3.3.2. Other Factors Significantly Influencing Progression-Free Survival

##### Tumor Size

An amount of 30/107 patients (28%) with pT1 had an event in comparison to 41/93 patients (44.1%; (HR 2.32 [1.4–3.86]; *p*-value: < 0.001). In particular, significantly more lymphatic node metastases were seen in the pT2 group when compared to the pT1 group (16/106 (15.1%) versus 25/92 (27.2%); (HR 2.44 [1.25–4.78]; *p*-value: 0.01). In addition, significantly more distant metastases were seen in cases of pT2 when compared to pT1 (6/106 (5.7%) versus 12/92/13%); HR 3.81 [1.25–11.62]; *p*-value: 0.02).

#### 3.3.3. Quality of Life

##### EORTC QLQ-C30 Global Health Status

When comparing both groups, significantly higher values were seen in the non-irradiated group six months after surgery (mean 61.89 vs. 53.99; *p* = 0.02). When comparing the means of all scores except global health status and financial difficulties, those differences were significant in favor of the non-irradiated group after six months (mean 80.08 vs. 68.48; *p* < 0.001) and after three years (mean 79 vs. 70.85; *p* = 0.03).

##### EORTC QLQ-C30 Physical Functioning and Pain

For physical functioning, no significant differences between groups could be detected (all *p* > 0.05). In contrast, patients after irradiation reported significantly higher values for pain after six months (mean 45.31 vs. 27.32; *p* < 0.001) and after one year (mean 37.3 vs. 25.28; *p* = 0.02).

##### EORTC QLQ-C30 with H&N 35

Additionally, including the H&N 35 module, “pain” was reported to be significantly higher in the group after irradiation after six months (mean 33.12 vs. 16.18; *p* < 0.001), two years (mean 27.76 vs. 19.63; *p* = 0.04), as well as after three years (mean 28.82 vs. 17.47; *p* = 0.01, Figure 4). Irradiated patients had significantly higher values in terms of impaired swallowing, speech, as well as teeth-related problems after six months (swallowing: mean 28.44 vs. 16.82; *p* < 0.001, speech-related: mean 30.98 vs. 20.64; *p* = 0.01, teeth-related: mean 40.6 vs. 21.87; *p* < 0.001) and after three years (swallowing: mean 31.02 vs. 17.43; *p* < 0.001, speech-related: mean 32.12 vs. 20.95; *p* = 0.03, teeth-related: mean 43.37 vs. 21.88, Figure 4).

In addition, the irradiated patients reported a significantly more restricted mouth opening after six months (mean: 51.23 vs. 28.25; *p* < 0.001), after one year (mean: 37.67 vs. 23.68; *p* = 0.02), and after two years (mean: 42.17 vs. 26.18; *p* = 0.01, Figure 4).

## 4. Discussion

To the best of our knowledge, this is the first prospective comprehensive cohort study that divided a homogeneous patient collective with oral squamous cell carcinoma of the oral cavity (pT1/2) or oropharynx (pT1), together with pN1, cM0, and pR0 into an adjuvant irradiation arm and a non-irradiated arm and used overall survival, progression-free survival, as well as quality of life as clinical parameters with a mean follow-up time of 3.4 years.

To summarize, adjuvant radiation therapy (RT) does not positively affect survival within the first five years. However, based on our data and the very low post hoc estimated power of less than 20%, this influence cannot be ruled out. However, adjuvant RT counteracts the occurrence of lymph node metastases and local recurrences within the first five years. In addition, it has to be kept in mind that such RT substantially decreases quality of life parameters, such as pain, impaired swallowing, speech, and teeth-related problems, together with a restricted mouth opening for up to three years. Keeping these results in mind, it has to be questioned if adjuvant RT should be advocated for all patients with pT1/2 OSCC with pN1 with no other adverse pathological features, as stated by some authors [8,12].

Most studies on pN1 OSCC indicate that an adjuvant RT reduces the risk of regional recurrence. Barry et al. conducted a matched-pair analysis on 90 intermediate-risk OSCC patients (30 with pN1). They showed, especially in the pN1 subgroup, a significant improvement in locoregional tumor control in the RT group [20]. The same results were seen in the study of Liu et al., who found differences through lower recurrence rates when applying adjuvant RT [21]. In an extensive US database analysis, Shrime et al. found a benefit of adjuvant RT in terms of survival among patients with pT2 pN1 primary tongue tumors and the floor of the mouth only [22]. Though, in this study, recurrence was not captured, details regarding RT dose and schedule, adjuvant RCT, and pathological information, such as ENE, resection margin, and vascular space/perineural invasion, were not available [22]. Another recent retrospective registry database analysis indicated that, in pT1/2 pN1-OSCC, adjuvant radiation therapy seems to benefit overall survival in females, patients > 64 years, pT2 tumors, and those with non-tongue cancer [13]. However, most studies could not detect differences in overall survival that follow our data [20,21,23].

It is known that patients treated for head and neck cancer via primary surgery and adjuvant RT show low quality of life scores, and the side effects of treatment should be outweighed by potentially improved cure rates [23]. It was reported that about six patients would rather endure the toxicity of radiotherapy in order to merit from prevention of one locoregional failure if the results were indiscriminately applied to all intermediate-risk patients [20]. Here, the general quality of life declines during and after treatment, but it seems to recover to the baseline levels after one year. Adversely, physical function scores related to saliva and swallowing seem to remain persistently low [20,24]. Bekiroglu et al. also detected a significant deficit in the physical scores following radiotherapy, even if direct matching of patients in TMN stage and margin status was not conducted. After one year and two years, substantial significant different and clinically relevant impairments were seen in terms of chewing, taste, and saliva (all potentially deriving from xerostomia) in the irradiated group [23]. We used the EORTC QLQ with H&N 35 module for quality-of-life assessment, a frequently reported head and neck cancer-specific instrument available in multiple languages. As an analog to the present findings, other authors reported a significant worsening of xerostomia at the end of radiation treatment with no observed improvements after one or even two years [25,26].

In addition to the underpowered analyses of the primary research parameter, several other potential biases are worth mentioning. As the study started in 2009, risk factors, such as close surgical margins (<5 mm), as well as depth of infiltration (≥10 mm; see changes in the 8th Edition of the American Joint Committee on Cancer (AJCC)), were not included in the protocol and, therefore, not documented in the present study [16]. As these parameters are known to have a relevant impact on prognosis after treatment, this constitutes a relevant bias. However, this represents a general problem in long-term prospective oncological studies, since more recent findings cannot be included in the ongoing investigation. When this trial was designed, the issue of human papillomavirus (HPV)-related oropharyngeal cancer (OPC) was not yet considered. The number of HPV-related OPC varies geographically in North America, where the prevalence is as high as 80%. In Europe, HPV prevalence differs between Northern Europe and Southern Europe, with significantly more “classic” cancers due to nicotine and alcohol consumption in the South. In Germany, approximately 30% of OPC are p16 positive. With only 29.7% of OPC within the study population and the relatively low HPV prevalence, the influence of p16-positive OPC in this population might be negligible. However, with this study, this issue remains unclear [27,28,29].

Even if randomized prospective clinical trials (RCTs) are the gold standard to generate evidence for comparing treatment efficacy, well designed observational studies might yield similar results [30]. However, a comprehensive cohort design cannot exclude risks, such as selection bias and systematic differences. For example, significantly more patients with pT2 and G > 2 received irradiation than pT1 and G1/G2. However, randomization might fail in study arms with enormously different therapy regimes, such as radiation therapy versus no radiation therapy, due to the “preference effect” (of the patient and/or physician) [16,31]. Another study bias is the different therapeutic regimes between the 27 study centers (“performance bias”), i.e., in terms of resection margins and lymph node management. Hence, we decided on stratification by the surgical treatment recommendations of the DÖSAK to improve internal validity [16,17].

## 5. Conclusions

When recommending adjuvant radiation therapy, the irradiation’s additional morbidity and consequent adverse effects must be communicated to the patient. Especially in the case of pT1/pT2, pN1, and M0-OSCC without other adverse pathological factors, the anticipated advantage in time to progression must be balanced against the radiation-induced toxicity. Therefore, in those cases, adjuvant radiation therapy should be considered individually to balance affected patients’ safety needs and quality of life.

## Figures and Tables

**Figure 1 cancers-15-01833-f001:**
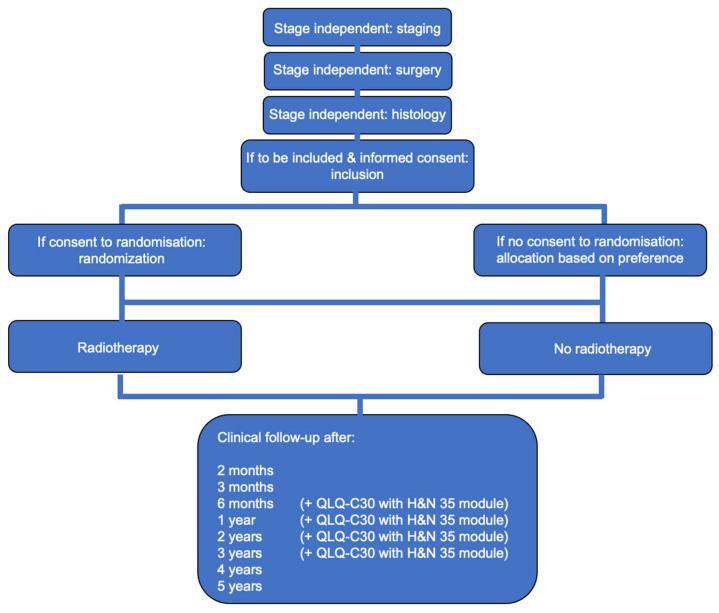
Flow-chart of the study.

**Figure 2 cancers-15-01833-f002:**
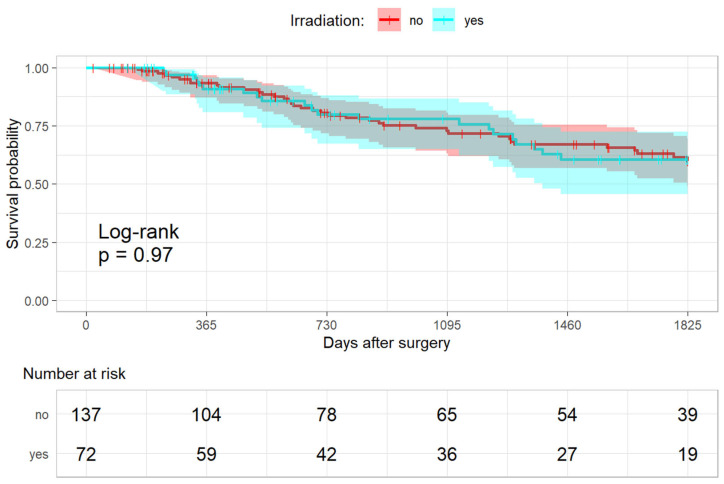
Kaplan-Meier analysis for overall survival.

**Figure 3 cancers-15-01833-f003:**
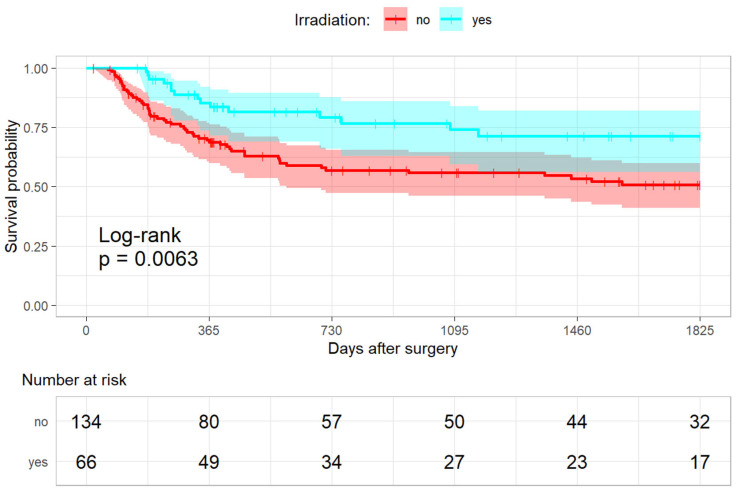
Kaplan-Meier analysis for the progression-free survival.

**Figure 4 cancers-15-01833-f004:**
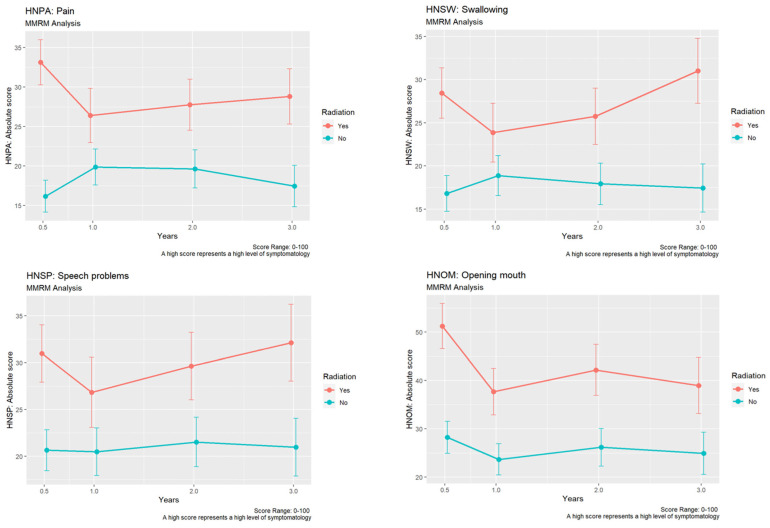
Development of differences in “pain”, “swallowing”, “speech problems”, and “opening mouth” using EORTC QLQ-C30 H&N 35 module between the non-irradiated and the irradiated group.

**Table 1 cancers-15-01833-t001:** Patient and treatment characteristics.

Variables		No Irradiation	Irradiation	Total	*p*-Value
** *Location* **					
	pre-canine	20 (14.6%)	10 (13.9%)	30 (14.4%)	0.71
	post-canine	70 (51.1%)	40 (55.5%)	110 (52.6%)	
	oropharyngeal	42 (30.7%)	20 (27.8%)	62 (29.7%)	
	(missing)	5 (3.6%)	2 (2.8%)	7 (3.3%)	
** *Side* **					
	left	61 (44.5%)	35 (49%)	96 (46%)	0.943
	right	76 (55.5%)	37 (51%)	113 (54%)	
** *Midline involvement* **					
	no	109 (79.6%)	57 (79.2%)	166 (79.4%)	>0.999
	yes	26 (19%)	14 (19.4%)	40 (19.2%)	
	(missing)	2 (1.4%)	1 (1.4%)	3 (1.4%)	
** *Type of reconstruction* **					
	Microvascular	61 (44.5%)	33 (45.8%)	94 (45%)	0.939
	Pedicled flap	8 (5.8%)	4 (5.6%)	12 (5.7%)	
	local	58 (42.3%)	27 (37.5%)	85 (40.7%)	
	(missing)	10 (7.3%)	8 (11.1%)	18 (8.6%)	
** *T classification* **					
	pT1	81 (59%)	29 (40.3%)	110 (52.6%)	0.014
	pT2	56 (41%)	43 (59.7%)	99 (47.4%)	
** *Grading* **					
	G1	4 (2.9%)	2 (2.8%)	6 (2.9%)	0.029
	G2	110 (80.3%)	46 (63.9%)	156 (74.6%)	
	G3	23 (16.8%)	24 (33.3%)	47 (22.5%)	
** *Neck level* **					
	1	27 (19.7%)	17 (23.6%)	44 (21%)	0.588
	2	80 (58.4%)	38 (52.8%)	118 (56.4%)	
	3	13 (9.5%)	11 (15.3%)	24 (11.3%)	
	4	2 (1.5%)	0 (0%)	2 (0.9%)	
	5	1 (0.7%)	0 (0%)	1 (0.4%)	
	(missing)	14 (10.2%)	6 (8.3%)	21 (10%)	

## Data Availability

Additional data can be obtained from the corresponding author.

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
