# Peer review of "Adjuvant Radiotherapy in Patients with Squamous Cell Carcinoma of the Oral Cavity or Oropharynx and Solitary Ipsilateral Lymph Node Metastasis (pN1)—A Prospective Multicentric Cohort Study"

_cancers, 2023, doi:10.3390/cancers15061833_

Round 1

Reviewer 1 Report

This is a multicenter prospective study. The majority of patients were treated in an observational study. A few patients were also randomized. It was investigated whether adjuvant RT is necessary or follow-up is sufficient for pT1/2 p N1 oral cavity carcinomas and pT1 p N1 oropharynx. No difference in OS was detected. However, the rate of local recurrences and lymph node metastases (DFS) was significantly worse when radiotherapy was omitted. In contrast, quality of life was significantly worse after radiation.

This is a relevant and clinically interesting question. In clinical practice, the question of radiotherapy in N1 situation is still a frequent unclear decision. Therefore, investigations are important in this field and thus the authors' results are relevant and the study is interesting for the readers. The  findings are new and may be helpful in further decision making of treatment choice of pT1/2 p N1 oral cavity carcinoma and in pT1 p N1 oropharyngeal carcinoma patients.

Overall, the study also has several limitations.

Revisions:

-Accurate information on patient characteristics is lacking. Exact tumor location in oral cavity or oropharynx, TNM information and exact information on applied radiation is missing for example. Such data, for both treatment groups, are very important for the interpretation and should be added, e.g. as a table.

-The Radiation is insufficiently described:

-The 59.4 Gy indicated were applied to the former tumor region and the affected LK region as shown in the text. However, were elective LK -levels also irradiated/what dose.

-It would also be interesting to know which elective levels were irradiated (were there study guidelines) and whether the irradiation was unilateral or bilateral.

All these parameters can influence the QOL and also the outcome and therefore more detailed information would be helpful for the interpretation.

-According to the data, the patients were treated at 27 centers.  If there were no uniform specifications for radiation therapy, this would strongly influence the results due to the high inhomogeneities to be expected.

-There is no precise information of how many oropharyngeal carcinomas were treated and how they were distributed between the two arms. Apparently, no information on HPV is available due to the time of implementation of the trial. However, since HPV positive patients develop significantly fewer recurrences this could have influenced the OS and DFS data. This is another limitation that needs to be addressed.

Author Response

Detailed Response to Reviewers

Adjuvant radiotherapy in patients with squamous cell carcinoma of the 
oral cavity or oropharynx and solitary ipsilateral lymph node metastasis 
(pN1) – a prospective multicentric cohort study
 (cancers-2242488).

Changes within the manuscript are highlighted in red.

First, we would like to thank the reviewers for their time and effort in helping us improve our manuscript. This is highly appreciated, and we did our best to address all valuable remarks.

Reply to Reviewer #1, detailed comments:

  1. Concern of the reviewer: “Accurate information on patient characteristics is lacking. Exact tumor location in oral cavity or oropharynx, TNM information and exact information on applied radiation is missing for example. Such data, for both treatment groups, are very important for the interpretation and should be added, e.g. as a table.”

Our response: We agree, and we added the respective information. See table 1. For information on applied radiation, please see our response #2.

  1. concern of the reviewer: “The Radiation is insufficiently described:

- The 59.4 Gy indicated were applied to the former tumor region and the affected LK region as shown in the text. However, were elective LK -levels also irradiated/what dose.

- It would also be interesting to know which elective levels were irradiated (were there study guidelines) and whether the irradiation was unilateral or bilateral.

All these parameters can influence the QOL and also the outcome and therefore more detailed information would be helpful for the interpretation.Ӊ۬

Our response: The authors appreciate this comment. The section “Adjuvant radiation therapy” was revised accordingly.

  1. concern of the reviewer: “According to the data, the patients were treated at 27 centers.  If there were no uniform specifications for radiation therapy, this would strongly influence the results due to the high inhomogeneities to be expected.”

Our response: Thank you for this comment. The authors refer to the revised section “Adjuvant radiation therapy”. The study protocol contained uniform specifications for trial centers on prescription, recording, and reporting the external beam radiation therapy.

  1. concern of the reviewer: “There is no precise information of how many oropharyngeal carcinomas were treated and how they were distributed between the two arms.“

Our response: This information was added (see table 1).

  1. concern of the reviewer: „ Apparently, no information on HPV is available due to the time of implementation of the trial. However, since HPV positive patients develop significantly fewer recurrences this could have influenced the OS and DFS data. This is another limitation that needs to be addressed.“

Our response: The authors agree with the reviewer. However, at the time when this trial was designed, the issue of HPV-related oropharyngeal cancer (OPC) was not yet considered. The number of HPV-related OPC varies geographically in North America, where the prevalence is as high as 80%. In Europe, HPV prevalence differs between Northern Europe and Southern Europe, with significantly more “classic” cancers due to nicotine and alcohol consumption in the South. In Germany, approximately 30% of OPC are p16 positive. With 29.7% of OPC cases within the study population, this relevant bias is now addressed in the discussion section.

Reviewer 2 Report

Major Comments:

-          This is a well performed, and written study.

Minor comments:

-          Line 232 should be reworded – e.g. There was no significant difference…

-          Line 249 - Consider changing “One hundred twenty-five” to the numerical form for ease of reading.

-          Figure 2 – Explain what the shading indicates.

-          Figure 3 – The labels and keys should be a little bigger for ease of reading.

Author Response

Reply to Reviewer #2, detailed comments:

  1. concern of the reviewer: “Line 232 should be reworded – e.g. There was no significant difference…”

Our response: We agreed and reworded the sentence.

  1. concern of the reviewer: “Line 249 - Consider changing “One hundred twenty-five” to the numerical form for ease of reading.”

Our response: Revision was done

  1. concern of the reviewer: “Figure 2 – Explain what the shading indicates.”

Our response: The shades indicate the “exponential” Greenwood 95%-confidence interval, which is based on the log (-log(survival)) transformation. This information was added to the manuscript.

  1. concern of the reviewer: “Figure 3 – The labels and keys should be a little bigger for ease of reading.”

Our response: Figures were adapted.

Round 2

Reviewer 1 Report

All points have been appropriately edited and the manuscript is now complete and understandable. Thus, it is suitable for publication.

Author Response

Thank you very much!